# Attosecond electron–spin dynamics in Xe 4d photoionization

Shiyang Zhong [1✉], Jimmy Vinbladh[2], David Busto [1], Richard J. Squibb[3], Marcus Isinger[1], Lana Neoričić [1], Hugo Laurell [1], Robin Weissenbilder[1], Cord L. Arnold[1], Raimund Feifel[3], Jan Marcus Dahlström[1], Göran Wendin[4], Mathieu Gisselbrecht [1], Eva Lindroth [2] & Anne L'Huillier [1]

The photoionization of xenon atoms in the 70–100 eV range reveals several fascinating physical phenomena such as a giant resonance induced by the dynamic rearrangement of the electron cloud after photon absorption, an anomalous branching ratio between intermediate $Xe^+$ states separated by the spin-orbit interaction and multiple Auger decay processes. These phenomena have been studied in the past, using in particular synchrotron radiation, but without access to real-time dynamics. Here, we study the dynamics of Xe 4d photo-ionization on its natural time scale combining attosecond interferometry and coincidence spectroscopy. A time-frequency analysis of the involved transitions allows us to identify two interfering ionization mechanisms: the broad giant dipole resonance with a fast decay time less than 50 as, and a narrow resonance at threshold induced by spin-flip transitions, with much longer decay times of several hundred as. Our results provide insight into the complex electron-spin dynamics of photo-induced phenomena.

[1] Department of Physics, Lund University, P.O. Box 118, Lund SE-221 00, Sweden. [2] Department of Physics, Stockholm University, AlbaNova University Center, Stockholm SE-106 91, Sweden. [3] Department of Physics, University of Gothenburg, Origovägen 6B, Gothenburg SE-412 96, Sweden. [4] Department of Microtechnology and Nanoscience—MC2, Chalmers University of Technology, Gothenburg SE-412 96, Sweden. ✉email: shiyang.zhong@fysik.lth.se

The absorption of X-rays by matter has been used since more than a century ago to understand the structure of its fundamental constituents[1]. An X-ray photon absorbed by an atom triggers multiple electron dynamics. The emission of an electron from an inner shell is accompanied by ultrafast rearrangement of the electronic cloud, which simultaneously modifies the potential seen by the electron, sometimes leading to resonances in the emission spectrum. An outer-shell electron may fill an inner hole, while another electron is emitted, a process called Auger decay[2]. Finally, the electron spin may be affected by the magnetic field induced by the ultrafast orbital motion, giving rise to spin flip, which is forbidden for purely electric dipole transitions. All of this complex hole or electron motion occurs on a rapid time scale, in the attosecond (1 as $= 10^{-18}$ s) range.

The interaction of xenon atoms with photons in the 70–100 eV range illustrates many aspects of the electron dynamics sketched above. Collective many-electron effects in the $4d$ shell[3–6] lead to a broad "giant dipole" resonance in the photoionization cross-section, which is maximum at 100 eV[7,8]. Photoionization is accompanied by Auger decay, involving the $5s$ and $5p$ shells, leading to the formation of $Xe^{2+}$ ions[9] (see Fig. 1). Relativistic (spin-orbit) effects can be observed at threshold (in the 70–75 eV region), with, in particular, an anomalous branching ratio between the $^2D_{5/2}$ and $^2D_{3/2}$ final states of the ion, separated by a spin-orbit splitting of 2 eV[10–12].

Attosecond pulses produced by high-order harmonic generation in gases[13,14] enable measuring ultrafast electron dynamics, as shown in a series of seminal experiments[15–22]. Temporal information is obtained by pump/probe techniques combining attosecond pulses and a synchronized laser field. The reconstruction of attosecond beating by interference of two-photon transition (RABBIT) technique[23], based on interferometry, allows the determination of the photoionization spectral amplitude in the complex plane. The temporal dynamics is then obtained by Fourier transform or more generally by time-frequency analysis[16,24]. This technique has been successfully used to measure photoionization time delays, due to electron propagation in the potential following the absorption of an extreme ultraviolet (XUV) photon. Most of these studies[25–28], however, have concentrated on relatively simple systems, ionized from the valence shells.

In this work, we present measurements of photoionization time delays in the Xe $4d$ shell for different ionic states $4d^{-1}(^2D_{5/2})$ and $4d^{-1}(^2D_{3/2})$, denoted $4d_{5/2}$ and $4d_{3/2}$ in the following (see Fig. 1b). Auger-photoelectron coincidence spectroscopy is used to disentangle electrons from different photoionization and decay channels[29]. The RABBIT interferometric technique allows the extraction of a phase, or a time (or group) delay, from the

photoelectron spectra. At high photon energy (between 80 and 100 eV), both $4d_{5/2}$ and $4d_{3/2}$ photoelectrons are emitted with the same positive time delay. Close to the $4d$-ionization threshold (75–80 eV), the measured time delays differ by more than 100 as. Supported by relativistic random phase approximation (RRPA) theoretical calculations[30], we show that this difference is due to the interference of the broad giant dipole resonance with a narrow threshold resonance due to relativistic spin-orbit effects.

## Results

The experiments were performed with attosecond pulse trains generated in neon by a femtosecond Ti:sapphire laser system, covering a spectral range from the $4d$ ionization threshold to the maximum of the giant dipole resonance (see Methods for details). A small fraction of the infrared (IR) laser beam was used as a probe with a variable time delay. The XUV and IR pulses were focused into Xe gas and the created electrons were detected by an electron spectrometer.

Photoionization to different ionic states followed by Auger decay produces a complex electron spectrum, with two sets of photoelectrons separated by 2 eV (see Fig. 1). Single Auger decay from $Xe^+$ (e.g., $4d_{5/2}$) to $Xe^{2+}$ (e.g., $5s^{-1}5p^{-1}$) leads to electrons at kinetic energies equal to the difference between intermediate and final state energies, spanning from 8.3 eV to 36.4 eV[31] and thus overlapping with the photoelectrons ionized by 75–100 eV photons[32]. Figure 2 shows XUV-only (a) and XUV+IR (b) two-dimensional coincidence maps. For a given final state of $Xe^{2+}$, Auger electrons detected in coincidence with photoelectrons contribute to a stripe with discrete spots related to absorption of different harmonics (with odd orders 53 to 63 in the figure), or absorption of harmonics and absorption or emission of an IR photon (sidebands 54 to 62). In addition, weak signals due to absorption or emission of an IR photon by the Auger electron, are observed (see, e.g., the difference between the blue and red curves in Fig. 2e at 9.8 eV). This coincidence technique requires long acquisition times, but allows us to disentangle unambiguously the $4d_{5/2}$ and $4d_{3/2}$ photoelectrons by the energy of the Auger electron.

Each sideband arises from the interference between two quantum paths as illustrated at the top of Fig. 2(d). The sideband signal oscillates as a function of the delay $\tau$ between the attosecond pulse train and the probe IR field, according to,

$$I_{SB} = A + B\cos(2\omega\tau - \phi), \quad (1)$$

where $A$ and $B$ are constants, $\omega$ is the IR frequency and $\phi$ is a phase offset, which can be extracted by fitting with a cosine function. The phase offset $\phi$ divided by the oscillation frequency ($2\omega$) can be written as the sum of two delays, $\tau_{XUV} + \tau_A$. The first one is the group delay of the attosecond pulses, while the second, called atomic time delay, arises from the two-photon ionization process. As shown in previous work[33,34] and as discussed in more details in the Supplementary Fig. 1, the variation of the atomic time delay $\tau_A$, as a function of XUV photon energy or between two spin-orbit split final states, reflects, to a large extent, one-photon ionization dynamics. To remove the influence of $\tau_{XUV}$ in our time delay measurements, we alternate experiments in Xe and Ne, and measure the time delay difference. Atomic time delays in Ne $2p$ can be measured and calculated with good accuracy. They are very small in the energy range considered[33], so that the time delay differences between Xe and Ne are, to a very good approximation, absolute time delays in Xe (see Supplementary Fig. 2). The sidebands corresponding to the same photoelectron but different Auger final state are found to oscillate in phase within our error bar, which allows us to average the time delays over the different Auger decay channels.

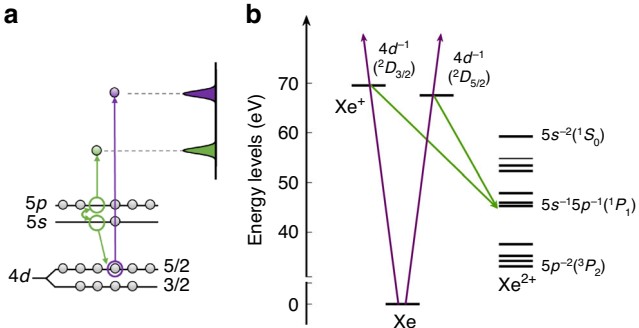

**Fig. 1 Excitation scheme. a** Schematic illustration of Xe $4d$ photoionization (violet) and Auger decay processes (green) after absorption of XUV radiation. **b** Xe energy diagram showing the $Xe^+$ intermediate and $Xe^{2+}$ final states involved.

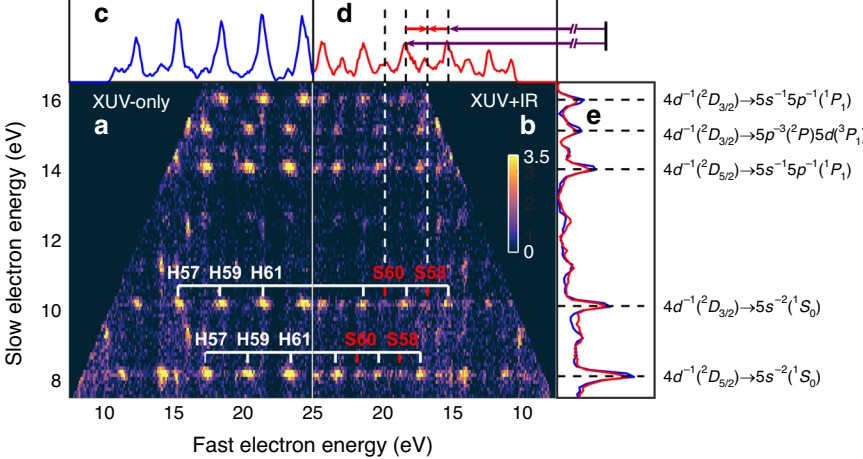

**Fig. 2 Two-electron coincidence map.** The coincidence map using **a** XUV field only and **b** XUV+IR fields. The number of measured two-electron pairs is indicated by the color code. The spots with constant slow-electron energy and variable fast-electron energy correspond to photoelectrons created by absorption of consecutive harmonics (labeled as H57-H61 in (a) as an example) and a given Auger electron (e.g., $4d^{-1}(^2D_{3/2}) \rightarrow 5s^{-2}(^1S_0)$). Photoelectrons corresponding to absorption or emission of an additional IR photon (sidebands, labeled as S58 and S60 as an example) appear in **b**. The projection on the fast electron energy axis **c** and **d** is the sum of the signal with slow electron energy from 10 eV to 10.4 eV, i.e., the photoelectrons in coincidence with $4d^{-1}(^2D_{3/2}) \rightarrow 5s^{-2}(^1S_0)$ Auger electrons. The projection on the slow electron energy axis **e** shows the sum of the signal for the different Auger processes indicated on the right, with (red) and without (blue) IR field. A RABBIT energy scheme is indicated at the top of **d**.

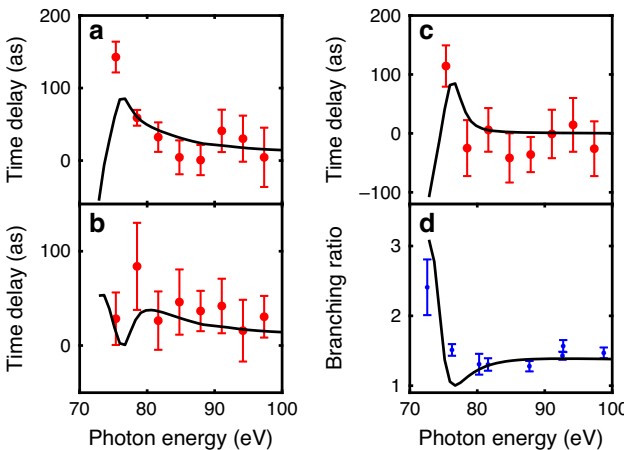

**Fig. 3 Atomic time delays and branching ratio.** Absolute atomic time delays for **a** $4d_{3/2}$ and **b** $4d_{5/2}$; **c** Difference between these delays. The experimental data are given in red dots. The black lines indicate the results of two-photon RRPA calculations. The estimation of the error bars is the standard error of the weighted mean, detailed in Methods section. In **d**, the calculated branching ratio of the $4d_{5/2}$ over $4d_{3/2}$ cross sections is shown in black. The experimental data in blue dots is from ref. [37].

Figure 3 presents the absolute atomic delays for $4d_{5/2}$ (a) and $4d_{3/2}$ (b) as a function of photon energy. We also present theoretical calculations obtained by solving the Dirac equation in RRPA (see Methods). At high photon energy (>80 eV), both $4d_{5/2}$ and $4d_{3/2}$ exhibit very similar positive time delay, about 40 as at 80 eV, slightly decreasing with energy. At low photon energy (<80 eV), the delays show a rapid variation with energy, opposite for the two final states. Theory and experiment are in good agreement at high photon energy. Although the agreement in the threshold region is not as good, the main features of the experiment are reproduced. Our theoretical calculations are in excellent agreement with the predictions of Mandal and coworkers[35,36].

Figure 3c shows the difference between $4d_{3/2}$ and $4d_{5/2}$ time delays. This difference is close to zero at high energy but jumps to

more than 100 as at 75 eV. Unfortunately, we could not reliably extract time delays below 75 eV, due to the low cross section and the overlap with double Auger electrons below 5 eV kinetic energy. The RRPA calculation predicts a strong decrease of the time delay difference towards the threshold. It also shows a strong deviation of the branching ratio between $4d_{5/2}$ and $4d_{3/2}$ cross sections from the statistical prediction in the same energy range, reproducing well experimental results[37,38] (see Fig. 3d).

## Discussion
To understand the underlying physics behind the variation of the time delays, we examine the behavior of the RRPA transition matrix elements involved in Xe $4d$ single photon ionization. In the energy range considered in this work, photoionization is dominated by the transitions from the $4d$ shell to continuum $f$ states, which we denote $4d \rightarrow \epsilon f$ in the following. The contribution from $4d \rightarrow \epsilon p$ transitions is one order of magnitude smaller in this energy region, as shown in the Supplementary Fig. 3. The asymptotic phase for a given channel is the sum of the Coulomb phase and a phase due to the short-range potential. The Coulomb phase is removed in the phases displayed in Fig. 4, as well as in the calculation of the time delays, in order to focus on the short range effects (see Supplementary Fig. 1). Figure 4a shows that photoionization is dominated by $4d_{3/2} \rightarrow \epsilon f_{5/2}$ and $4d_{5/2} \rightarrow \epsilon f_{7/2}$, especially at high photon energy, in the region of the giant dipole resonance. In the threshold region, the $4d_{5/2} \rightarrow \epsilon f_{5/2}$ channel contributes significantly. This transition is accompanied by a spin flip, which points out the role of the spin-orbit interaction. The phases and time delays for the three channels (Fig. 4b,c) coincide above 80 eV photon energy, showing the first half of a $\pi$ phase variation across the giant dipole resonance with a time delay of ~40 as. Below 80 eV, the three quantities plotted in Fig. 4a–c show a strong, oscillating, channel dependence, indicating a quantum interference phenomenon.

The dynamics behind this effect can be unraveled by calculating the Wigner representation[24,39], defined as the Fourier transform of the auto-correlation function of the transition matrix elements, $D_i(E)$, $i$ denoting the channel, and $E$ the electron

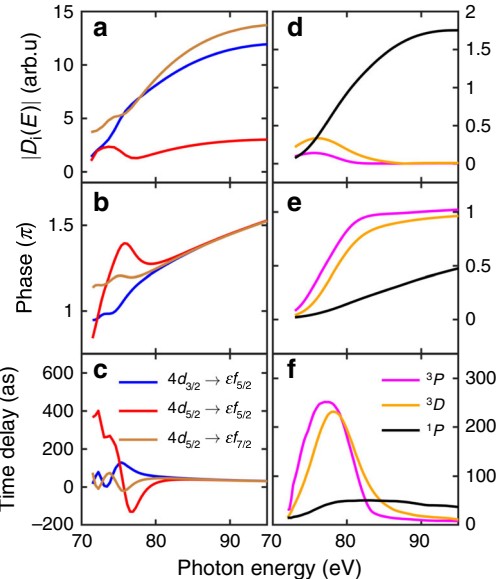

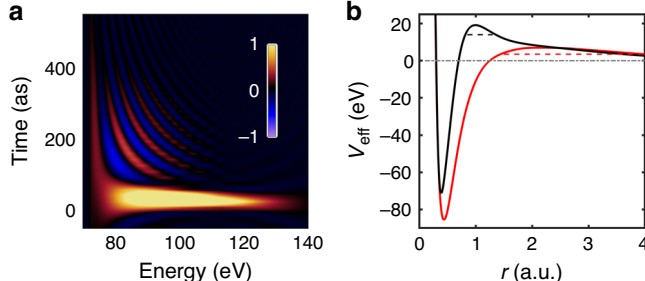

**Fig. 4 Transition matrix elements. a, d** Modulus, **b, e** phase and **c, f** time delay of the transition matrix element $D_i(E)$ as a function of photon energy for the coupled channels (**a–c**) $4d_{3/2} \rightarrow \epsilon f_{5/2}$ (blue), $4d_{5/2} \rightarrow \epsilon f_{5/2}$ (red), $4d_{5/2} \rightarrow \epsilon f_{7/2}$ (brown) and eigenchannels (**d–f**) $^1P$ (black), $^3P$ (magenta), $^3D$ (orange), extracted from ref. [12]. We exclude the Coulomb phase-shift in **b, e** and **c, f**. A $\pi$ phase shift has been added to the phase of $4d_{5/2} \rightarrow \epsilon f_{5/2}$ for better comparison. The time delay is the energy derivative of the phase.

**Fig. 5 Wigner representation and effective potentials. a** Wigner representation $W(E, t)$ for the $4d_{5/2} \rightarrow \epsilon f_{5/2}$ channel. The amplitude is indicated by the color code on the right. **b** Illustration of one-electron potentials for the $^1S \rightarrow {}^3P$, $^3D$ transitions (red) and $^1S \rightarrow {}^1P$ (black). Dashed lines suggest possible electron trajectories in the two cases.

kinetic energy.

$$W(E, t) = \frac{1}{2\pi} \int_{-\infty}^{\infty} d\epsilon D_i\left(E + \frac{\epsilon}{2}\right) D_i^*\left(E - \frac{\epsilon}{2}\right) e^{-\frac{i\epsilon t}{\hbar}}, \quad (2)$$

where $\hbar$ is the reduced Planck constant. The results are shown in Fig. 5a for the $4d_{5/2} \rightarrow \epsilon f_{5/2}$ channel (see Supplementary Fig. 4, for the other two channels). All three channels show similar features. (i) A broad resonance with a maximum around 100 eV and a short decay of a few tens of attosecond, which can be interpreted as the giant dipole resonance; (ii) A sharp resonance at low energy, around 75 eV, with a long decay of a few hundreds of attoseconds; (iii) Interferences between these resonances, leading to rapid oscillations of the Wigner distribution.

To interpret the sharp spectral feature at 75 eV, we utilize the theoretical analysis performed in the seminal work of Cheng and Johnson[12]. Using a multichannel quantum defect theory approach[40], results obtained within RRPA, similar to those of the present work, were analyzed and eigenchannel solutions were extracted. These eigenchannel solutions are completely decoupled from each other and can be used as a basis to describe coupled-channel transitions. They are labeled using the closest corresponding LS-coupled channel [$(d^9f)$ and $(d^9p)^1P$, $^3P$, $^3D$], where in all these cases it is only the J = 1 states that can be populated from the xenon ground state by one-photon absorption. Neglecting the weak contribution of the $4d \rightarrow \epsilon p$ transitions, the $4d \rightarrow \epsilon f$ transitions are superpositions of $(d^9f)$ $^1P$, $^3P$ and $^3D$ eigenchannels (In the following, we drop the $d^9f$ label).

Figure 4d–f present the modulus, phase and time delay of these three eigenchannels. The behavior of the three curves is much simpler than those in Fig. 4a–c. For each eigenchannel, a single resonance feature can be identified, with a peak for the modulus and time delay and a $\pi$ phase variation across the resonance. While the broad feature, maximum at 100 eV, obviously represents the giant dipole resonance ($^1S \rightarrow {}^1P$), the narrow peaks at 75 and 76 eV exist because of the spin-orbit interaction that

enables singlet to triplet mixing. The maximum of the time delay varies from a few tens ($^1P$) to a few hundreds ($^3D$ and $^3P$) of attoseconds, in agreement with the results in Fig. 5a.

The difference in time delays can be further interpreted by examining the effective potential experienced by the escaping $f$ photoelectron. We represent in Fig. 5b a mean-field average potential (red), as well as the potential modified by $^1S \rightarrow {}^1P$ dipole polarization (screening) effects (black), which are included in the random phase approximation with exchange (RPAE) approach. These effects lead to an effective high and narrow potential barrier and therefore to a broad resonance, with a maximum at high energy, and a short decay time (see black dashed line). In contrast, an electron emitted in the triplet channels does not feel these dipole polarization effects and sees essentially the potential indicated in red, with a relatively low barrier only due to angular momentum and a long decay time (red dashed line). The time delay is directly related to the resonance lifetime, being equal to it at the maximum of the resonance[41]. Figure 5b even suggests that the increase of the temporal width of the broad resonance in Fig. 5a towards low energy might be due to the influence of the long tail of the screened potential (black).

The rapid variation of the amplitude, phase and delays of the three $4d \rightarrow \epsilon f$ channels (Fig. 4a–c) at threshold can therefore be interpreted as a quantum interference effect between the "direct" dipole-allowed $^1S \rightarrow {}^1P$ transition and the spin-orbit-induced $^1S \rightarrow {}^3P$, $^3D$ transitions, which have similar amplitudes in this region. This interference explains the difference in time delays for $4d_{3/2}$ and $4d_{5/2}$, as well as the anomalous branching ratio (Fig. 3c, d).

In conclusion, we have measured photoionization time delays in Xe using attosecond interferometry, giving us high temporal resolution, and coincidence spectroscopy, which allows us to avoid spectral congestion and to obtain a high spectral resolution. These time delays are positive and similar for the two spin-orbit split Xe$^+$ states over a large energy range (up to 100 eV photon energy), except at threshold (75 eV) where they differ by 100 as. With the help of RRPA calculations for one- and two-photon ionization, we attribute this difference to the interference of several channels coupled by the spin-orbit interaction. A time-frequency analysis of the dominant transition matrix elements, allows us to unravel two main ionization processes, with very different time and energy scales: the broad giant dipole resonance, dominated by the $^1S$ to $^1P$ transition and the narrow resonances due to the $^1S$ to $^3P$ and $^3D$ transitions, which are enabled by the spin-orbit interaction. While the former takes place over a few tens of attoseconds, the latter, involving spin flip, occurs over several hundreds of attoseconds. Our experimental approach, which adds temporal information to

traditional spectral studies, provides increased understanding of the complex electron dynamics taking place in Xe 4d photoionization. Finally, shape resonances are ubiquitous in nature and the methods developed in this work should be useful to investigate the electronic properties of a variety of molecular systems (see recent work in $N_2$[42] and $CH_3I$[43]).

## Methods

**Experimental method**. The experiments were performed with 40-fs long pulses, centered at 800 nm with 1-kHz repetition rate from a Ti:sapphire femtosecond laser system. The laser was focused into a 6-mm long gas cell filled with Ne to generate high-order harmonics. A 200-nm thick Zr filter was used to filter out the infrared (IR) and most of the harmonics below the Xe 4d threshold (67.5 eV for $4d_{5/2}$). The filtered harmonics thus span the 4d ionization region from the threshold to the maximum of the giant dipole resonance (100 eV). A small fraction (30%) of the IR beam, split-off before generation, is used as a probe with a variable time delay. The XUV and IR pulses were focused in an effusive Xe gas jet. The electrons were detected by a magnetic bottle electron spectrometer, which combines high collection efficiency and high spectral resolution up to $E/\Delta E \sim 80$.

**Data analysis**. Each data point in Fig. 3 is the arithmetic mean weighted with the uncertainty estimated from the cosine fitting to Eq. (1). In each measurement, we average the time delays of electron pairs corresponding to the same photoelectron but different Auger decay. For $N$ measurements yielding $N$ data points: $\tau_1, \tau_2, \ldots, \tau_N$ with corresponding uncertainties: $\sigma_1, \sigma_2, \ldots, \sigma_N$, the weighted average can be calculated as:

$$\overline{\tau} = \frac{\sum_{i=1}^{N} w_i \tau_i}{\sum_{i=1}^{N} w_i}, \tag{3}$$

where $w_i = 1/\sigma_i^2$ is the weight. The uncertainty for each measurement is estimated from the fit of the RABBIT oscillation to a cosine function. The uncertainty on the time delay difference, $\tau_A - \tau_B$, can be expressed as:

$$\sigma = \sqrt{\sigma_A^2 + \sigma_B^2}. \tag{4}$$

The error bars of the experimental results indicate the standard error of the weighted mean and can be calculated as:

$$\sigma_{\overline{\tau}} = \sqrt{\frac{N}{(N-1)\omega_s^2} \sum \omega_i^2 (\tau_i - \overline{\tau})^2}, \tag{5}$$

where $\omega_s = \Sigma \omega_i$.

**Theoretical method**. Theoretical calculations consisted in calculating one-photon and two-photon matrix elements within lowest-order perturbation theory for the radiation fields, using wavefunctions obtained by solving the Dirac equation, and including electron correlation effects within the RPAE for one-photon XUV photoionizaton.

The complex-valued two-photon matrix elements[44,45] are calculated following the procedure described in[46,47] for the non-relativistic case. Briefly, the absorption of one ionizing photon is treated within the RPAE approximation and a perturbed wave function is calculated. Exterior complex scaling is used in order to be able to use a finite numerical box. The two-photon matrix element is dominated by a one-photon dipole matrix element between the intermediate perturbed wave function and the final continuum state[48]. The integration is performed numerically out to a distance far outside the atomic core, but within the unscaled region, while the last part of the integral is carried out using analytical Coulomb waves along the imaginary radial axis. The amplitude and phase shift of these Coulomb waves are determined from the numerical solutions for the perturbed wave function and for the final state describing a free electron within the potential of the remaining ion. The numerical stability is monitored by comparison of different "break points" between the numerical and analytical descriptions.

A few adjustments have to be made in the relativistic case: Relativistic RPAE is used to obtain the perturbed wave function after absorption of one photon and the Dirac Hamiltonian to determine the phase shift of the final state. The Coulomb solution for the large component approaches the non-relativistic solution in the asymptotic region, albeit with relativistically adjusted parameters. For a given energy, the small component is easily obtained from the large one[30].

## Data availability

The data that support this study are available online https://doi.org/10.5878/rhak-nd96 at Swedish National Data Service. Source data are provided with this paper.

## Code availability

The codes used in this study are available from the corresponding author upon reasonable request.

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

## Acknowledgements
The authors acknowledge support from the Swedish Research Council, the European Research Council (advanced grant PALP-339253), the Knut and Alice Wallenberg Foundation and Olle Engkvist's Foundation.

## Author contributions
S.Z., D.B., R.J.S., M.I., L.N., H.L., R.W. and C.L.A. performed the experiment. R.J.S and R.F. provided part of the experimental setup. J.V., J.M.D. and E.L. performed RRPA calculations. S.Z., D.B., J.M.D., G.W., M.G., E.L. and A.L. worked on the analysis and the theoretical interpretation. S.Z. and A.L. wrote the main part of the manuscript. All authors gave feedback on the manuscript.

## Funding

## Competing Interests
The authors declare no competing interests.
