## [Peer Review File · Nature Communications]

REVIEWER COMMENTS

Reviewer #1 (Remarks to the Author):

In their manuscript, Zhong, et.al present a study of attosecond photoionization dynamics in Xenon atoms near the 4d ionization threshold in the energy range of 70-100 eV. They perform the experiments using coincidence spectroscopy and present theoretical calculations to support their findings. They perform a time-frequency analysis and identify two interfering ionization mechanisms, one involving the giant dipole resonance and the other involving a spin-flip transition.

This study uses a combination of the well-established techniques of RABBITT and coincidence spectroscopy to obtain new insights into attosecond dynamics near the very interesting giant resonance in Xenon. The analysis and results presented are novel and thorough and will be interesting to a broad audience. These results will definitely influence and encourage further exploration of attosecond photoionization dynamics, especially in molecules. The statistical analysis presented is valid and sufficient information is provided in the manuscript and supplementary information (SI) to allow reproduction of the experimental results and the analysis. I find this manuscript suitable for Nature Communications and recommend publication. I have a few suggestions that may help improve the manuscript:

1. At the beginning of page 5 it is abruptly mentioned that the photoionization is dominated by 4d \rightarrow ϵf transitions. While it is clear that the authors are referring to the continuum channel of the photoelectron, a non-specialist may find this confusing. It will be helpful if it is clearly mentioned what the 4d \rightarrow ϵf transitions correspond to.
2. On the same point as above, the authors mention both in the manuscript and the SI that the 4d \rightarrow ϵf transitions dominate. While there is a discussion of the effect of 4d \rightarrow ϵp transitions on the dynamics in the SI, there is really no evidence provided as to why the authors state this. If theoretical calculations indicate the dominance of the 4d \rightarrow ϵf transitions, the authors should consider providing some quantitative details.
3. The introduction can be improved. While the authors have attempted to provide a broadly accessible introduction, over simplification such as referring to the spin of an electron as its rotational motion should be avoided.

Niranjan Shivaram

Reviewer #2 (Remarks to the Author):

In this manuscript the authors presents the result of a combined experimental and theoretical study of the dynamics of the highly correlated motion of multiple electrons after formation of a core hole. The chosen case of atomic xenon is a prime example of ultrafast electronic rearrangement, especially the shape resonance at ~ 100 eV has been targeted by numerous studies. The team has applied their well established RABBIT method but combined it with electron-electron coincidence

detection, thus being able to associate each Auger electron with the photon energy that has led to the corresponding inner-shell hole. This way, the atomic phases / time-delays that are extracted from the delay-dependent sideband intensities can be plotted as function of the photon energy. For the two fine-structure components, the result exhibits a non-zero but constant atomic phase difference within the shape resonance but also a deviation from that behavior close to the 4d ionization threshold. Admittedly, only a single data point of fig. 4c) at 75 eV photon energy supports this finding, however, with a solid 3-sigma significance. Thanks to the attending theoretical treatment, the authors are able to explain this behaviour as a consequence of the interference of two ionization channels, one of them being associated with a spin-flip transition.

The paper demonstrates the amount of insight that can be gained into extremely fast electronic events, if sophisticated time-metrology is combined with contemporary theoretical treatment. The authors have succeeded in unveiling the physical mechanism behind the observed dynamics, thus underlining once more the value of xenon as a laboratory for ultrafast multi-particle correlation. I recommend publication in Nature Communications in the current form.

Reviewer #3 (Remarks to the Author):

The present paper addresses nicely using relativistic Dirac equation simulations attosecond time delays in photoionization of a 4d electron in Xe. In particular Fig 2 shows convincingly the RABBIT scheme as sidebands induced by the IR pulse. The main difficulty in the interpretation is the influence of the time delay T_A due to the two separate two photon transitions. The calculation of the two photon matrix element via the Dirac equation involves a resonant and a nonresonant part of a Greens, function. As shown in nonrelativistic time delays by HJ Woerner (PRL 117,093001(2016)) from simulations by Kawai, PRA 75,063402(2007) and Ivanov, PRA 86,063422(2012), the nonresonant continuum-continuum transitions are important contributions to the atomic time delay. No mention is made of the relativistic effects on these resonant and nonresonant 2-photon transitions. How these are calculated and compared would be instructive. Finally the transition selection rules, $\Delta J=0,1,2$ to the individual atomic states are not identified and would be useful for proper understanding of the physics. Proper attention to these points will make the paper more complete and very acceptable.

Response to Reviews

Reviewer #1:

In their manuscript, Zhong, et.al present a study of attosecond photoionization dynamics in Xenon atoms near the 4d ionization threshold in the energy range of 70-100 eV. They perform the experiments using coincidence spectroscopy and present theoretical calculations to support their findings. They perform a time-frequency analysis and identify two interfering ionization mechanisms, one involving the giant dipole resonance and the other involving a spin-flip transition.

This study uses a combination of the well-established techniques of RABBITT and coincidence spectroscopy to obtain new insights into attosecond dynamics near the very interesting giant resonance in Xenon. The analysis and results presented are novel and thorough and will be interesting to a broad audience. These results will definitely influence and encourage further exploration of attosecond photoionization dynamics, especially in molecules. The statistical analysis presented is valid and sufficient information is provided in the manuscript and supplementary information (SI) to allow reproduction of the experimental results and the analysis. I find this manuscript suitable for Nature Communications and recommend publication.

We sincerely thank the encouraging comments from the reviewer.

I have a few suggestions that may help improve the manuscript:

1. At the beginning of page 5 it is abruptly mentioned that the photoionization is dominated by $4d \rightarrow \epsilon f$ transitions. While it is clear that the authors are referring to the continuum channel of the photoelectron, a non-specialist may find this confusing. It will be helpful if it is clearly mentioned what the $4d \rightarrow \epsilon f$ transitions correspond to.

We specified that the transition is from 4d to continuum following the suggestion:

"photoionization is dominated by the transitions from the 4d shell to continuum f states, which we denote $4d \rightarrow \epsilon f$ in the following."

2. On the same point as above, the authors mention both in the manuscript and the SI that the $4d \rightarrow \epsilon f$ transitions dominate. While there is a discussion of the effect of $4d \rightarrow \epsilon p$ transitions on the dynamics in the SI, there is really no evidence provided as to why the authors state this. If theoretical calculations indicate the dominance of the $4d \rightarrow \epsilon f$ transitions, the authors should consider providing some quantitative details.

In the Supplementary Material (Figure S3), the modulus of transition matrix element of $4d \rightarrow \epsilon f$ and $4d \rightarrow \epsilon p$ transitions, which is proportional to the square root of photoionization cross section, is given.

We revised the sentence in the manuscript:

"The contribution from $4d \rightarrow \epsilon p$ transitions is one order of magnitude smaller in this energy region, as shown in the SM (Fig.~S3)"

3. The introduction can be improved. While the authors have attempted to provide a broadly accessible introduction, over simplification such as referring to the spin of an electron as its rotational motion should be avoided.

We have tried to avoid over simplification, and removed our reference to the electron spin as its rotational motion. The new sentence in the revised manuscript reads:

“Finally, the electron spin may be affected by the magnetic field induced by the ultrafast orbital motion, giving rise to spin flip, which is forbidden for purely electric dipole transitions.”

Furthermore, we have changed “An X-ray penetrating inside an atom” into “An X-ray photon absorbed by an atom”.

Writing a broadly accessible introduction is not an easy task and we thank the referee for his/her critical comments!

Reviewer #2 (Remarks to the Author):

In this manuscript the authors presents the result of a combined experimental and theoretical study of the dynamics of the highly correlated motion of multiple electrons after formation of a core hole. The chosen case of atomic xenon is a prime example of ultrafast electronic rearrangement, especially the shape resonance at $\sim 100\text{V}$ has been targeted by numerous studies. The team has applied their well established RABBIT method but combined it with electron-electron coincidence detection, thus being able to associate each Auger electron with the photon energy that has led to the corresponding inner-shell hole. This way, the atomic phases / time-delays that are extracted from the delay-dependent sideband intensities can be plotted as function of the photon energy. For the two fine-structure components, the result exhibits a non-zero but constant atomic phase difference within the shape resonance but also a deviation from that behavior close to the 4d ionization threshold. Admittedly,

only a single data point of fig. 4c) at 75 eV photon energy supports this finding, however, with a solid 3-sigma significance. Thanks to the attending theoretical treatment, the authors are able to explain this behaviour as a consequence of the interference of two ionization channels, one of them being associated with a spin-flip transition.

The paper demonstrates the amount of insight that can be gained into extremely fast electronic events, if sophisticated time-metrology is combined with contemporary theoretical treatment. The authors have succeeded in unveiling the physical mechanism behind the observed dynamics, thus underlining once more the value of xenon as a laboratory for ultrafast multi-particle correlation.

I recommend publication in Nature Communications in the current form.

We sincerely thank the encouraging comments from the reviewer.

Reviewer #3 (Remarks to the Author):

The present paper addresses nicely using relativistic Dirac equation simulations attosecond time delays in photoionization of a 4d electron in Xe. In particular Fig 2 shows convincingly the RABBIT scheme as sidebands induced by the IR pulse. The main difficulty in the interpretation is the influence of the time delay τ_A due to the two separate two-photon transitions. The calculation of the two-photon matrix element via the Dirac equation involves a resonant and a nonresonant part of a Greens function. As shown in nonrelativistic time delays by HJ Woerner (PRL 117,093001(2016)) from simulations by Kawai, PRA 75,063402(2007) and Ivanov, PRA 86,063422(2012), the nonresonant continuum-continuum transitions are important contributions to the atomic time delay. No mention is made of the relativistic effects on these resonant and nonresonant 2-photon transitions. How these are calculated and compared would be instructive.

We thank the referee for this comment. We have extended the “theoretical method” in the Methods section to give more explanation on how the complex two-photon matrix elements (with the real and imaginary parts called resonant and nonresonant in the article from Kawai and Bandrauk, Phys. Rev. A, 2007) are calculated.

“The complex-valued two-photon matrix elements [KawaiPRA2007,DahlstromJPB2012] are calculated following the procedure described in [DahlstromPRA2012,DahlstromJPB2014] for the non-relativistic case. Briefly, the absorption of one ionizing photon is treated within the RPAE approximation and a perturbed wave function is calculated. Exterior complex scaling is used in order to be able to use a finite numerical box. The two-photon matrix element is dominated by a one-photon dipole matrix element between the intermediate perturbed wave function and the final continuum state [VinbladhPRA2019]. The integration is performed numerically out to a distance far outside the atomic core, but within the unscaled region, while the last part of the integral is carried out using analytical Coulomb waves along the imaginary radial axis. The amplitude and phase shift of these Coulomb waves are determined from the numerical solutions for the perturbed wave function and for the final state describing a free electron within the potential of the remaining ion. The numerical stability is monitored by comparison of different “break points” between the numerical and analytical descriptions.

A few adjustments have to be made in the relativistic case: Relativistic RPAE is used to obtain the perturbed wave function after absorption of one photon and the Dirac Hamiltonian to determine the phase shift of the final state. The Coulomb solution for the large component approaches the non-relativistic solution in the asymptotic region, albeit with relativistically-adjusted parameters. For a given energy, the small component is easily obtained from the large one [Vinbladhthesis,Vinbladh2019].”

The referee also asks about a comparison between relativistic and nonrelativistic calculations for the two-photon matrix elements. Because of the number of channels involved, which differs in the two cases, we show here a comparison for the atomic, Wigner and continuum-continuum delays (section I of the supplementary information) in order to separate the effect of one-photon absorption (Wigner delay, grey) and continuum-continuum transition (red) and to compare with the two-photon delay (blue). The figure on the left is a non-relativistic calculation, while that on the right is relativistic. (The dashed grey line on the right corresponds to the red line in Fig. S1, which goes to lower energies.)

As shown in these graphs, relativistic effects mainly affect the threshold region, and absorption of the first XUV photon (Wigner delay). The delay due to the continuum-continuum transition is essentially the same in the relativistic and non-relativistic case (except close to 10 eV kinetic energy). In fact, the non-relativistic “universal” continuum-continuum delay calculated analytically in the asymptotic approximation [DahlstromCP2013], shown in black in Fig. S1, is very close to the relativistic calculation.

We have clarified this point in the SM, when commenting Fig. S1. We have added (see in blue in the revised manuscript):

“The atomic delay τ_A is here calculated in a relativistic framework. In the considered energy range, the difference compared to the Wigner delay (also calculated relativistically) is close to the τ_{cc} obtained non-relativistically using an analytical, universal, i.e. atom and channel-independent formula [DahlstromCP2013]. Generally speaking, relativistic effects are significant in a rather narrow spectral region close to the 4d threshold, and affect mainly the first (XUV) photon absorption.”

We have also clarified the caption of Fig. S1 (see in blue).

Finally, we notice that all of the references indicated by the referee are about molecular systems. Inspired by this, we have added a sentence in the conclusion (last sentence) opening towards more complex systems, and allowing us to cite related recent work. “Finally, shape resonances are ubiquitous in nature and the methods developed in this work should be useful to investigate the electronic properties of a variety of molecular systems (see recent work in N_2 [ref] and CH_3I [ref]).”

Finally the transition selection rules, $\Delta J=0,1,2$ to the individual atomic states are not identified and would be useful for proper understanding of the physics.

Since the ground state of xenon is of $J=0$ symmetry, only $J=1$ symmetries can be reached with one photon and, as the referee says, $J=0,1,2$ with two-photons, (with a general light pulse). The resonances with 1P, 3P and 3D symmetries discussed in the manuscript all refer to odd parity states that can be reached by one photon. They are thus all of $J=1$ symmetry.

To clarify this, we revised the sentence in p.5 in the manuscript:

“They are labelled using the closest corresponding LS-coupled channel [(d9f) and (d9p) 1P, 3P, 3D], where in all these cases it is only the $J=1$ states that can be populated from the xenon ground state by one-photon absorption.”

Proper attention to these points will make the paper more complete and very acceptable.

We thank the comments that help us strengthen the quality of the work and we hope that the referee find our article now appropriate for publication in Nature Communications.

REVIEWERS' COMMENTS

Reviewer #3 (Remarks to the Author):

After looking at the revised version, the authors are to be congratulated for a complete and quite acceptable new version ready for publication.

Response to the reviewer

Reviewer #3 (Remarks to the Author):

After looking at the revised version, the authors are to be congratulated for a complete and quite acceptable new version ready for publication.

We sincerely thank the reviewer for the precious comments that help us to improve our manuscript.